# AN ANALYTICAL SOLUTION TO GAUSS-NEWTON LOSS FOR DIRECT IMAGE ALIGNMENT

**Sergei Solonets**[1,2,*]     **Daniil Sinitsyn**[1,2,*]
**Lukas von Stumberg**[3,†]     **Nikita Araslanov**[1,2]     **Daniel Cremers**[1,2]

[1] Technical University of Munich
[2] Munich Center for Machine Learning
[3] Valve Software

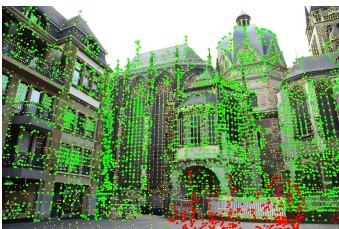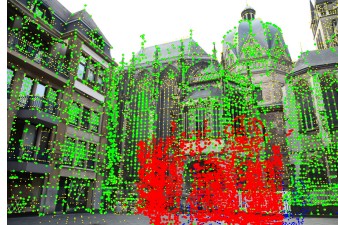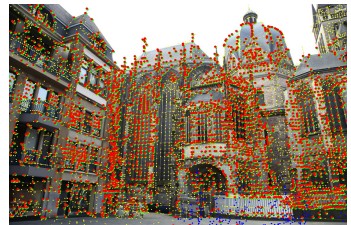

Figure 1: **Direct image alignment** is a technique for aligning scenes based on image intensities. Recent learning-based methods seek to improve its success rate by increasing the convergence basin. We derive an analytical solution to the core idea of such methods, the Gauss-Newton loss, enabling fine-grained control over the basin of convergence. As a result, we can successfully align two scenes despite a highly imprecise initialization. From left to right, the example above illustrates the convergence of the reprojected keypoints (red) to the ground truth (green) by optimizing the $\mathbf{SE(3)}$ camera pose with our analytical solution. The blue color is the re-projection in the first iteration.

## ABSTRACT

Direct image alignment is a widely used technique for relative 6DoF pose estimation between two images, but its accuracy strongly depends on pose initialization. Therefore, recent end-to-end frameworks increase the convergence basin of the learned feature descriptors with special training objectives, such as the Gauss-Newton loss. However, the training data may exhibit bias toward a specific type of motion and pose initialization, thus limiting the generalization of these methods. In this work, we derive a closed-form solution to the expected optimum of the Gauss-Newton loss. The solution is agnostic to the underlying feature representation and allows us to dynamically adjust the basin of convergence according to our assumptions about the uncertainty in the current estimates. These properties allow for effective control over the convergence in the alignment process. Despite using self-supervised feature embeddings, our solution achieves compelling accuracy *w. r. t.* the state-of-the-art direct image alignment methods trained end-to-end with pose supervision, and demonstrates improved robustness to pose initialization. Our analytical solution exposes some inherent limitations of end-to-end learning with the Gauss-Newton loss, and establishes an intriguing connection between direct image alignment and feature-matching approaches.

## 1 INTRODUCTION

Visual localization refers to estimating the camera pose of a query image *w. r. t.* a reference image where the underlying 3D structure (*e. g.* a point cloud) is available. Traditionally, solutions to visual localization primarily relied on estimating correspondences between 2D features in the query image and 3D features in the reference point cloud (Liu et al., 2017; Sarlin et al., 2019; Sattler et al., 2017;

---

[*]Equal contribution. Correspondence: {s.solonets,d.sinitsyn}@tum.de.
[†]Work done while at TU Munich.
 Project code: https://github.com/tum-vision/gn_loss_analytical.

Svarm et al., 2017; Toft et al., 2018). Challenging this approach, recent deep learning frameworks implement direct image alignment by re-projecting the 3D points onto the feature map (Sarlin et al., 2021; von Stumberg et al., 2020a;b). Using end-to-end training, deep networks learn dense feature maps suitable for regressing the pose between an image and a point cloud. In analogy to photometric image alignment (Delaunoy and Pollefeys, 2014; Engel et al., 2016), this family of methods is referred to as *featuremetric* image alignment (von Stumberg et al., 2020b).

The feature representation learned by featuremetric image alignment is not only discriminative, but is also spatially smooth for improved convergence (von Stumberg et al., 2020a). A training process enforces the smoothness either explicitly through a specific loss, such as the Gauss-Newton loss (von Stumberg et al., 2020a), or implicitly, by backpropagating through an optimization algorithm, such as Levenberg-Marquardt (Sarlin et al., 2021). In both cases, the resulting feature map embeds the bias of the initial poses in the training data. This may lead to poor generalization, since the training poses can differ substantially from the test scenario. We here take a less bias-prone approach: we use generic feature descriptors (*e. g.* obtained with self-supervision) and instead control the smoothness of the feature map dynamically at test time.

This work investigates the connection between feature descriptor networks and featuremetric image alignment. The *main contribution* is the analytical (closed-form) solution to the Gauss-Newton loss. On the one hand, this leads to a novel technique, which can utilize *any* feature descriptor to generate a dense feature map suitable for direct image alignment. Importantly, it allows us to dynamically adjust the smoothness of the feature map, which effectively controls the trade-off between the basin of convergence and the alignment accuracy. We empirically verify our derivation using *self-supervised* feature descriptors, such as SuperPoint (DeTone et al., 2018), and demonstrate on-par or even superior alignment accuracy compared to *supervised* frameworks. On the other hand, the analysis of our closed-form solution reveals inherent limitations of feature learning with backpropagation through Gauss-Newton optimization: featuremetric alignment merely learns a form of interpolation between feature descriptors in the points of interest. Although we demonstrate this in the context of direct image alignment, a similar argument extends to other methods, even beyond computer vision, which backpropagate through Gauss-Newton or Levenberg-Marquardt optimization.

## 2 PRELIMINARIES

**Image Alignment.** Given two images (*reference* $I_r$ and *query* $I_q$) with known camera models and an overlapping field of view, and a 3D point cloud $\{\boldsymbol{p}^{(i)}\}$ in the coordinate system of $I_r$, the problem of *image alignment* is to estimate the relative camera pose $\boldsymbol{T} \in \mathbf{SE}(3)$.

**Gauss-Newton Optimization.** Given a set of functions $\{r^{(1)}(\boldsymbol{x}), ..., r^{(m)}(\boldsymbol{x})\}$ called *residuals*, *Gauss-Newton (GN) optimization* finds the parameters minimizing the sum of squared residuals:

$$f(\boldsymbol{x}) = \sum_i \|r^{(i)}(\boldsymbol{x})\|_2^2. \tag{1}$$

Each residual $r^{(i)}(\boldsymbol{x})$ could either be a vector or a single-valued function. In both scenarios, the total residual $r(\boldsymbol{x})$ is a stacked vector of $m$ residuals. The Gauss-Newton method seeks to minimize $f(\boldsymbol{x})$ by iteratively updating the parameter estimate $\tilde{\boldsymbol{x}}$. Each iteration linearizes the residuals around the current estimate and computes an update step $\Delta_{GN}(r(\tilde{\boldsymbol{x}}))$:

$$\Delta_{\mathrm{GN}}[r(\tilde{\boldsymbol{x}})] = (\boldsymbol{J}^{\mathsf{T}}\boldsymbol{J})^{-1}\,\boldsymbol{J}^{\mathsf{T}}r(\tilde{\boldsymbol{x}}), \quad \boldsymbol{J} := \frac{\partial r}{\partial \boldsymbol{x}}\Big|_{\boldsymbol{x}=\tilde{\boldsymbol{x}}}, \tag{2}$$

where $\boldsymbol{J}$ is the Jacobian matrix. The estimate $\tilde{\boldsymbol{x}}$ evolves until convergence as

$$\tilde{\boldsymbol{x}} \leftarrow \tilde{\boldsymbol{x}} \boxminus \Delta_{\mathrm{GN}}[r(\tilde{\boldsymbol{x}})]. \tag{3}$$

Here, the operator $\boxminus$ denotes a specific update procedure, which depends on the nature of the optimization space. For linear spaces, $\boxminus$ simplifies to the standard subtraction operation; in the context of a rigid-body transformation $\mathbf{SE}(3)$, operator $\boxminus$ applies a tangential update.

**Photometric and featuremetric image alignment.** Photometric image alignment estimates the relative 6DoF pose by minimizing the difference between pixel intensities of points in $I_r$ and the

corresponding points in $I_q$. Recalling $\boldsymbol{p}^{(i)}$ as a 3D point in the coordinate frame of the reference image, we seek a transformation $\boldsymbol{T} \in \mathbf{SE}(3)$ minimizing the following residuals:

$$r^{(i)}(\boldsymbol{T}) = I_r\big(\langle \boldsymbol{p}^{(i)} \rangle\big) - I_q\big(\langle \boldsymbol{T} \boldsymbol{p}^{(i)} \rangle\big). \tag{4}$$

Here, $\langle \cdot \rangle$ is a 2D projection operator of a 3D point onto the image plane. The projection implicitly uses the corresponding camera intrinsic parameters, assumed to be available for both $I_r$ and $I_q$. We omit them in the notation for clarity. Comprising rotation $\boldsymbol{R} \in \mathbf{SO}(3)$ and translation $\boldsymbol{t} \in \mathbb{R}^3$, the target transformation $\boldsymbol{T}$ is typically found by minimizing Eq. (4) with non-linear least-squares optimization, such as Gauss-Newton or Levenberg-Marquardt (LM) methods (Levenberg, 1944; Marquardt, 1963; Nocedal and Wright, 1999).

The success of photometric image alignment critically depends on a favorable initialization of the pose, especially in the conditions of varying illumination and occlusion. As a partial remedy, *featuremetric* image alignment (Tang and Tan, 2019) uses feature maps instead of pixel intensities. Obtained with deep learning, such feature maps have increased convergence basin compared to that derived from image intensities, which improves robustness to pose initialization.

## 3 Related Work

**Direct and indirect image alignment.** Estimating the relative camera pose from two images is a fundamental problem in computer vision, with applications in structure from motion (SfM) (Schönberger and Frahm, 2016), SLAM and relocalization. To address this problem, *indirect* (feature-based) approaches detect and match interest points (Bay et al., 2006; Lowe, 2004) in both images, and then estimate the pose using PnP (Persson and Nordberg, 2018) or by minimizing the reprojection errors (Triggs et al., 1999). In contrast, *direct* methods sidestep the matching process and minimize the photometric error instead (Horn and Jr., 1988; Irani and Anandan, 1999). This means that they can leverage the entire image (Kerl et al., 2013; Newcombe et al., 2011), or focus on pixels with sufficient gradient (Engel et al., 2014; 2016). The foundation behind these approaches is Lucas-Kanade tracking (Baker and Matthews, 2004; Lucas and Kanade, 1981). However, direct methods, which are central to this work, optimize for a 6DoF pose instead of individual pixel displacements (*i. e.* optical flow).

**Pose estimation with deep learning.** The advent of deep learning revitalized interest in improving pose estimation with deep networks. While some approaches are holistic (Jatavallabhula et al., 2020), there are broadly three categories of learning-based methods. *i) Fully end-to-end pose estimation methods* (Kendall et al., 2016; Ummenhofer et al., 2017; Zhou et al., 2017) directly regress pose estimates with deep neural networks, instead of test-time optimization. *ii) Learning-based indirect methods* (DeTone et al., 2018; Dusmanu et al., 2019; Revaud et al., 2019; Yi et al., 2016) replace handcrafted detectors and descriptors with deep representations in an indirect pipeline. Some approaches further extend the traditional way of obtaining correspondences. SuperGlue (Sarlin et al., 2020) learns feature matching with a graph neural network. LoFTR (Sun et al., 2021) directly regresses correspondences instead of relying on separate feature detection and matching. Similar to indirect methods, *iii) learning-based direct image alignment* enhances classical direct methods with deep features. This category is the most similar to our work and we discuss it in more detail next.

**Learned features for direct image alignment.** Previous work differs in their approach to model training and in the final task. For example, Czarnowski et al. (2017) leverage off-the-shelf CNN features to improve optical flow tracking. A number of methods (Han et al., 2018; Lv et al., 2019; Sarlin et al., 2021; Tang and Tan, 2019; Xu et al., 2021) train feature descriptors end-to-end with ground-truth poses and backpropagate the gradient through a non-linear optimization process. At test time, these methods employ a feature pyramid and refine the initial camera pose in a coarse-to-fine fashion using GN or LM optimization. In addition to the feature pyramid, some methods predict additional properties for image alignment, such as uncertainty (Xu et al., 2021) (Sarlin et al., 2021), Jacobians (Han et al., 2018), or optimization parameters (*e. g.* damping factors (Lv et al., 2019; Sarlin et al., 2021)). In the same spirit, our formulation leads to a continuous image pyramid, where each level of the pyramid can be generated on-the-fly based on the input level of uncertainty.

Another line of work (von Stumberg et al., 2020a;b) trains a deep network directly on the ground-truth pixel correspondences. GN-Net (von Stumberg et al., 2020a) minimizes two loss functions. The first is a contrastive loss (Schmidt et al., 2017) facilitating discriminative properties of the

feature representation. The second loss function accounts for a likely displacement in the initial correspondence, implemented by adding random 2D offsets to the ground-truth correspondences *at training time*. By learning to minimize the feature discrepancy after a Gauss-Newton step, the model learns to cope with the initially noisy estimates. We revisit this work in detail in the Sec. 4.

The learning-based approaches (*e. g.* (Germain et al., 2021; Sarlin et al., 2021; von Stumberg et al., 2020a)) achieve impressive accuracy of pose estimation. Nevertheless, they require pose supervision for training and struggle in scenarios of large-baseline localization. Indeed, GN-Net (von Stumberg et al., 2020a) and PixLoc (Sarlin et al., 2021) tend to exhibit a strong bias toward the noise assumptions of the training process, which cannot be easily reversed. For example, PixLoc trained on the CMU dataset exhibits a strong bias toward horizontal movement as illustrated in Appendix A.

Our work takes a different approach. We rely on existing feature descriptors obtained with self-supervision, which contribute no explicit motion bias to the alignment process. We next derive a closed-form solution to the Gauss-Newton loss (von Stumberg et al., 2020a) as a functional of a probability density governing the noise assumptions of the current pose estimate. This allows us to adjust the noise assumptions in the alignment process, thus effectively controlling the convergence basin at test time, much akin to the coarse-to-fine strategy, exemplified by Fig. 2.

## 4    THE GAUSS-NEWTON LOSS

In this section, we recap the GN-Net (von Stumberg et al., 2020a) and introduce the notation. GN-Net is a convolutional neural network $E(\,\cdot\,;\boldsymbol{\theta})$ trained on a sparse set of ground-truth correspondences. Given coordinates on the image plane, let $f : \mathbb{R} \times \mathbb{R} \to \mathbb{R}^d$ continuously map those coordinates to a descriptor from a feature grid $\mathbb{R}^{d \times w \times h}$ produced by $E(\,\cdot\,;\boldsymbol{\theta})$, *e. g.* using bilinear interpolation. We define $f_r := E(I_r; \boldsymbol{\theta})$ and $f_q := E(I_q; \boldsymbol{\theta})$ to denote feature representations of reference and query images. GN-Net learns parameters $\boldsymbol{\theta}$ to minimize the expected value of the following loss function,

$$\mathcal{L}_{\text{GN-Net}}(I_r, I_q) = \mathcal{L}_{\text{contrastive}}(f_r, f_q) + \mathcal{L}_{\text{GN}}(f_r, f_q),  \tag{5}$$

which comprises a contrastive loss, $\mathcal{L}_{\text{contrastive}}(\cdot, \cdot)$, and the Gauss-Newton loss, $\mathcal{L}_{\text{GN}}(\cdot, \cdot)$. The contrastive loss minimizes the distance between the features of two corresponding points while maximizing the distance between non-corresponding pairs (Schmidt et al., 2017). The contrastive loss facilitates spatially discriminative features, and its particular instantiation has little significance for the following discussion (*e. g.* GN-Net uses the triplet loss).

The Gauss-Newton loss $\mathcal{L}_{\text{GN}}$ ensures that the feature map is sufficiently smooth for direct image alignment, thus enlarging the convergence basin. GN-Net implements this by adding a random offset to the ground-truth correspondences and encouraging a single Gauss-Newton step to recover the original location. Let us formalize this process.

Given a ground-truth correspondence $(\boldsymbol{x}^{(i)}, \boldsymbol{y}^{(i)})$ between images $I_r$ and $I_q$, the assumption behind the Gauss-Newton loss is that the initial estimate $\tilde{\boldsymbol{x}}^{(i)}$ at test time falls in the vicinity of the ground truth $\boldsymbol{x}^{(i)}$, *i. e.* $\tilde{\boldsymbol{x}}^{(i)} = \boldsymbol{x}^{(i)} + \boldsymbol{\epsilon}$, where $\boldsymbol{\epsilon}$ follows some predefined distribution $p(\boldsymbol{\epsilon})$, such as a Gaussian with zero mean. At training time, the Gauss-Newton loss aims at recovering the ground-truth location $\boldsymbol{x}^{(i)}$ from a noisy initial location $\tilde{\boldsymbol{x}}^{(i)}$ by minimizing the residual *w. r. t.* $\boldsymbol{x}$:

$$r^{(i)}(\tilde{\boldsymbol{x}}^{(i)}) := f_r(\tilde{\boldsymbol{x}}^{(i)}) - f_q(\boldsymbol{y}^{(i)}),  \tag{6}$$

The original Gauss-Newton loss for one point $\boldsymbol{x}^{(i)}$, as introduced by von Stumberg et al. (2020a), is

$$\mathcal{L}_{\text{GN}_o}^{(i)}(f_r, f_q, \boldsymbol{\epsilon}) := \|\boldsymbol{J}(\boldsymbol{\epsilon} - \Delta_{\text{GN}}[r^{(i)}(\boldsymbol{x}^{(i)} + \boldsymbol{\epsilon})])\|_2^2 - \text{logdet}\,\boldsymbol{J}^{\mathsf{T}}\boldsymbol{J}.  \tag{7}$$

From a probabilistic standpoint (von Stumberg et al., 2020a), the loss balances between the accuracy of the Gauss-Newton step and the direction uncertainty. Here, $\boldsymbol{J}^{\mathsf{T}}\boldsymbol{J}$ represents the inverse covariance matrix propagated through the photometric residuals. The loss function corresponds to the negative log-likelihood of residuals distributed as $\mathcal{N}\left(0, (\mathbf{J}^{\mathsf{T}}\mathbf{J})^{-1}\right)$. In this work, we consider a variant of the Gauss-Newton loss, in which the covariance matrix is assumed to be identity. Thus, Eq. (7) becomes

$$\mathcal{L}_{\text{GN}}^{(i)}(f_r, f_q, \boldsymbol{\epsilon}) = \|\boldsymbol{\epsilon} - \Delta_{\text{GN}}[r^{(i)}(\boldsymbol{x}^{(i)} + \boldsymbol{\epsilon})]\|_2^2.  \tag{8}$$

Although this simplified version does not account for the trade-off between uncertainty and accuracy of the prediction, it admits a closed-form minimizer of the expected value, as we show in Sec. 5. We also find that the simplified version does not differ from the original one empirically in a significant way (see Appendix C). Henceforth, we will refer to the simplified version as the Gauss-Newton loss.

The training process calculates the Gauss-Newton loss stochastically by sampling $\epsilon$ from $p(\epsilon)$. Therefore, it minimizes a Monte-Carlo approximation to the expected value of $\mathcal{L}_{\text{GN}}^{(i)}$ summed over all ground-truth correspondences,

$$\mathcal{L}_{\text{GN}}(f_r, f_q; p) = \mathbb{E}_{\epsilon \sim p} \big[ \sum_i \mathcal{L}_{\text{GN}}^{(i)}(f_r, f_q, \epsilon) \big]. \tag{9}$$

Hereafter, we use the notation $\mathcal{L}_{\text{GN}}(f_r, f_q; p)$ to emphasize the dependence of the Gauss-Newton loss on the noise distribution $p$, and omit this parametarization otherwise to avoid clutter.

Note that the GN-Net's training stage addresses the problem of optical flow, not pose estimation. The underlying assumption is that accurate optical flow facilitates pose estimation, as each pixel will contribute to the final pose estimate. At test time, the pose is determined by solving the featuremetric image alignment problem (equivalent to Eq. (4)) using Gauss-Newton optimization in $\mathbf{SE}(3)$.

## 5 CLOSED-FORM GAUSS-NEWTON STEP

**Decoupling the contrastive and Gauss-Newton losses.** The contrastive loss provides a sparse constraint on the feature embeddings, since we can only use sparse ground-truth correspondences, the *interest points*, for supervision. By contrast, the Gauss-Newton loss enforces a pre-defined basin of convergence *around* each interest point (modeled by $p(\epsilon)$) with little effect on the feature descriptors in the interest points. This is because the residual in Eq. (6) between the corresponding interest points will be negligible, if the contrastive loss for those points is minimized. It follows that the contrastive and the Gauss-Newton loss essentially optimize over a *disjoint* set of feature locations. Therefore, we can decouple the Gauss-Newton loss from the joint optimization objective in Eq. (5). Let us formalize this reasoning. We aim to solve:

$$f_q^* = \arg\min_{f_q} \left[ \mathcal{L}_{\text{contrastive}}(f_r, f_q) + \mathcal{L}_{\text{GN}}(f_r, f_q) \right]. \tag{10}$$

We denote the values of $f_r$, $f_q$ in the interest points as $F_r$, $F_q$: $F_r^{(i)} := f_r(\boldsymbol{x}^{(i)})$, $F_q^{(i)} := f_q(\boldsymbol{y}^{(i)})$. The contrastive loss only depends on $F_r$ and $F_q$, while the Gauss-Newton loss depends on all the query feature values $f_q$ and the interest points in the reference map $F_r$. By eliminating unused parts, we introduce an equivalent problem:

$$\{F_q^*, f_q^*\} = \arg\min_{F_q, f_q} \left[ \mathcal{L}_{\text{contrastive}}(F_r, F_q) + \mathcal{L}_{\text{GN}}(F_r, f_q) \right]. \tag{11}$$

Next, we approximate $F_r$ with $F_q$ in the second term. This is permissible as the contrastive loss acts as a soft constraint, ensuring that $F_r$ and $F_q$ are close at the optimum of the joint loss function. This allows us to decouple the minimization problem as follows:

$$F_q^* = \arg\min_{F_q} \left[ \mathcal{L}_{\text{contrastive}}(F_r, F_q) + \min_{f_q} \mathcal{L}_{\text{GN}}(F_q, f_q) \right],$$
$$f_q^* = \arg\min_{f_q} \mathcal{L}_{\text{GN}}(F_q^*, f_q). \tag{12}$$

We denote

$$G(F_q; p) := \arg\min_{f_q} \mathcal{L}_{\text{GN}}(F_q, f_q; p),$$
$$\mathcal{L}_{\text{GN}}^*(F_q; p) := \min_{f_q} \mathcal{L}_{\text{GN}}(F_q, f_q; p) = \mathcal{L}_{\text{GN}}(F_q, G(F_q; p); p). \tag{13}$$

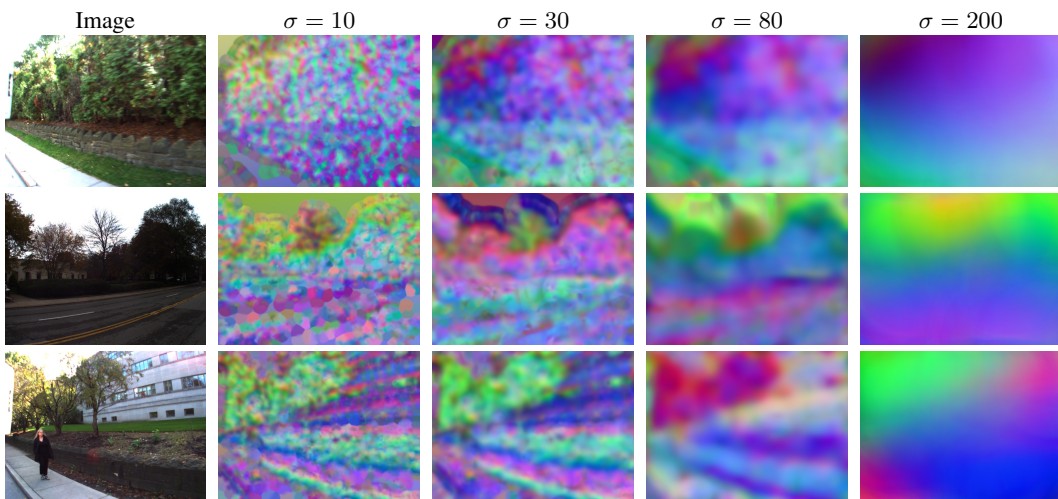

Figure 2: **Controlling the basin of convergence.** Image samples and the corresponding feature maps $\tilde{f}_q$ (PCA) for different values of $\sigma$ and isotropic Gaussian distribution $p$.

$G(F_q; p)$ is the optimal reconstruction of a feature map under sparse feature descriptors $F_q$ and noise density $p(\epsilon)$. $\mathcal{L}^*_{\text{GN}}(F_q; p)$ is the corresponding value of the Gauss-Newton loss under $F_q$. We will refer to $G(F_q; p)$ as $\tilde{f}_q$ to denote an optimal reconstruction under an arbitrary sparse input $F_q$. Note the difference to $f_q^*$, which is an optimal reconstruction under the joint loss, $i.\,e.\ G(F_q^*; p)$. Our main contribution, detailed shortly, is a closed-form solution for both $G(F_q; p)$ and $\mathcal{L}^*_{\text{GN}}(F_q; p)$. It reveals that the original problem in Eq. (10) actually depends only on the feature values in the interest points:

$$F_q^* = \arg\min_{F_q} \left[ \mathcal{L}_{\text{contrastive}}(F_r, F_q) + \mathcal{L}^*_{\text{GN}}(F_q; p) \right]. \tag{14}$$

This means that in training a feature extractor only the features of interest points matter, while the representation of other pixels approximates our analytical solution. As the empirical validation, we demonstrate compelling results in our experiments by using feature descriptors for the interest points from a self-supervised method, while employing our analytical solution for the remaining pixels.

**An analytical solution to the Gauss-Newton loss.** We propose to calculate the expectation of the Gauss-Newton loss in Eq. (9) as a functional of $f_q$, and analytically find its closed-form minimizer.

Consider the query image $I_q$ as a set $\{(\boldsymbol{x}^{(j)}, F_q^{(j)})\}$ containing locations of interest points $\boldsymbol{x}^{(j)}$ and the corresponding feature descriptors $F_q^{(j)} := f_q(\boldsymbol{x}^{(j)})$. The locations $\boldsymbol{x}^{(j)}$ can be extracted by an off-the-shelf feature detector (*e. g.* SuperPoint (DeTone et al., 2018)), while a network trained with the contrastive loss in Eq. (5) can produce the corresponding descriptors $F_q^{(j)}$.

Let us first re-write the expectation in Eq. (9) using Eq. (8) and Eq. (2) in the decoupled formulation of the Gauss-Newton loss (*cf.* Eq. (12)):

$$\mathcal{L}_{\text{GN}}(F_q, f_q; p) = \sum_j \int_\Omega \left\| \boldsymbol{\epsilon} - (\boldsymbol{J}^\mathsf{T}\boldsymbol{J})^{-1}\boldsymbol{J}^\mathsf{T}\left( f_q(\boldsymbol{x}^{(j)} + \boldsymbol{\epsilon}) - F_q^{(j)} \right) \right\|_2^2 p(\boldsymbol{\epsilon}) d\boldsymbol{\epsilon}. \tag{15}$$

Substituting $\boldsymbol{\epsilon} = \tilde{\boldsymbol{x}} - \boldsymbol{x}^{(j)}$, we obtain

$$\mathcal{L}_{\text{GN}}(F_q, f_q; p) = \sum_j \int_\Omega \left\| \tilde{\boldsymbol{x}} - (\boldsymbol{J}^\mathsf{T}\boldsymbol{J})^{-1}\boldsymbol{J}^\mathsf{T}\left( f_q(\tilde{\boldsymbol{x}}) - F_q^{(j)} \right) - \boldsymbol{x}^{(j)} \right\|_2^2 p(\tilde{\boldsymbol{x}} - \boldsymbol{x}^{(j)}) d\tilde{\boldsymbol{x}}. \tag{16}$$

We first consider all $\tilde{\boldsymbol{x}} \in \Omega$ independently and relax the problem by eliminating the constraint $\boldsymbol{J} = \frac{\partial f_q}{\partial \tilde{\boldsymbol{x}}}$. This allows us to derive analytical solutions for minimizers $\tilde{f}_q(\cdot)$ and $\tilde{J}_q(\cdot)$ as functions

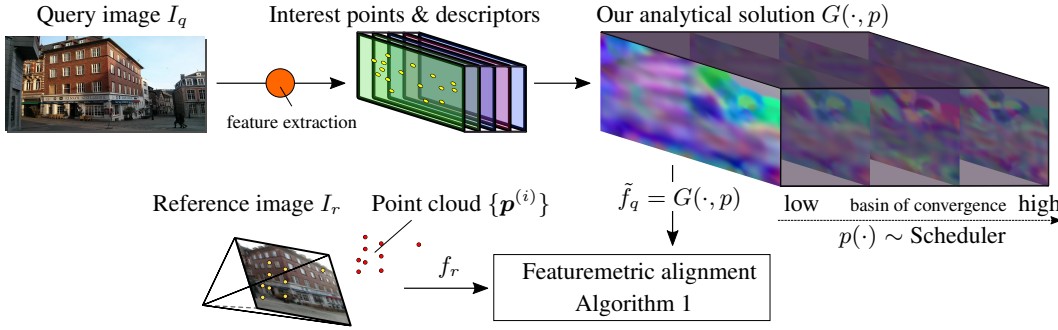

Figure 3: **Image alignment with continuous probabilistic feature pyramids.** Given reference image with a 3D point cloud and a query image paired with an initial coarse pose, we start by estimating interest points and their features on the query image, along with features from the projections of the 3D point cloud onto the reference image. Next, we perform featuremetric alignment between the 3D point features and derived analytical continuous image pyramid.

of $\tilde{x}$. We then observe that $\frac{\partial}{\partial \tilde{x}} \tilde{f}_q(\cdot)$ coincides with $\tilde{J}_q(\cdot)$ for a uniform distribution, which confirms that in this case, our derived solution is the solution to the original problem. As a side note, in the context of direct image alignment, it has been suggested that decoupling and predicting the Jacobian independently from the function value may yield superior convergence and results (Han et al., 2018). Appendix B provides the full derivation. Below, we summarize the closed-form minimizer to Eq. (16):

$$\tilde{f}_q(\tilde{\boldsymbol{x}}) = \tilde{J}_q(\tilde{\boldsymbol{x}})\big(\tilde{\boldsymbol{x}} - \mathbf{x}_m(\tilde{\boldsymbol{x}})\big) + \mathbf{y}_m(\tilde{\boldsymbol{x}}), \tag{17}$$

$$\tilde{J}_q(\tilde{\boldsymbol{x}}) = \big(\mathrm{Cov}_{\mathbf{xy}}(\tilde{\boldsymbol{x}})\mathrm{Cov}_{\mathbf{y}}(\tilde{\boldsymbol{x}})^{-1}\big)^{+}, \tag{18}$$

$$\mathbf{y}_m(\tilde{\boldsymbol{x}}) = \sum_j F_q^{(j)} p(\tilde{\boldsymbol{x}} - \boldsymbol{x}^{(j)}) / \sum_j p(\tilde{\boldsymbol{x}} - \boldsymbol{x}^{(j)}), \tag{19}$$

$$\mathbf{x}_m(\tilde{\boldsymbol{x}}) = \sum_j \boldsymbol{x}^{(j)} p(\tilde{\boldsymbol{x}} - \boldsymbol{x}^{(j)}) / \sum_j p(\tilde{\boldsymbol{x}} - \boldsymbol{x}^{(j)}), \tag{20}$$

$$\mathrm{Cov}_{\mathbf{xy}}(\tilde{\boldsymbol{x}}) = \sum_j \big(\boldsymbol{x}^{(j)} - \mathbf{x}_m(\tilde{\boldsymbol{x}})\big) \big(F_q^{(j)} - \mathbf{y}_m(\tilde{\boldsymbol{x}})\big)^{\mathsf{T}} p(\tilde{\boldsymbol{x}} - \boldsymbol{x}^{(j)}), \tag{21}$$

$$\mathrm{Cov}_{\mathbf{y}}(\tilde{\boldsymbol{x}}) = \sum_j \big(F_q^{(j)} - \mathbf{y}_m(\tilde{\boldsymbol{x}})\big) \big(F_q^{(j)} - \mathbf{y}_m(\tilde{\boldsymbol{x}})\big)^{\mathsf{T}} p(\tilde{\boldsymbol{x}} - \boldsymbol{x}^{(j)}), \tag{22}$$

where $(^{+})$ denotes the Moore–Penrose pseudo-inverse operator. The optimal point is $G(F_q; p) := \tilde{f}_q$ and the loss value at this point (*cf.* Eq. (13)) is

$$\mathcal{L}_{GN}^{*}(F_q) = \int_\Omega \frac{1}{2} \mathrm{Tr} \big[\mathrm{Cov}_{\mathbf{x}}(\tilde{\boldsymbol{x}}) - \mathrm{Cov}_{\mathbf{xy}}(\tilde{\boldsymbol{x}})\mathrm{Cov}_{\mathbf{y}}(\tilde{\boldsymbol{x}})^{-1}\mathrm{Cov}_{\mathbf{yx}}(\tilde{\boldsymbol{x}})\big] \, d\tilde{\boldsymbol{x}}. \tag{23}$$

Our derivation of the optimal embeddings $G(F_q; p)$ using fixed interest points $F_q$ provides an interesting insight into end-to-end learning frameworks. *Jointly* training both losses requires inverting a high-dimensional matrix $\mathrm{Cov}_{\mathbf{y}}(\tilde{\boldsymbol{x}})$ in Eq. (23), which poses high numerical instability and discontinuities. Since previous work (von Stumberg et al., 2020a) can be seen as stochastic approximations to our solution, this may explain the reported training instability and divergence in those works. Additionally, the form of $\tilde{f}_q$ in Eq. (17) suggests that end-to-end pipelines may not necessarily yield any sophisticated representation, as they merely interpolate between features in the interest points.

**Featuremetric Image Alignment.** As a case study, we employ the analytical form of $\tilde{f}_q$ and $\tilde{J}_q$ for featuremetric image alignment. Algorithm 1 and Fig. 3 provide an overview. Fig. 1 illustrates stages of direct alignment using our probabilistically reconstructed feature maps. We first extract the interest points $\{\boldsymbol{x}^{(j)}\}$ and the corresponding feature descriptors $\{F_q^{(j)}\}$ from $I_q$ to reconstruct $\tilde{f}_q$ and

**Require:** $I_r, I_q, \{\boldsymbol{p}^{(i)}\}, \boldsymbol{T}^{(0)}$
1: $\{\boldsymbol{x}^{(j)}\} \leftarrow \text{InterestPoints}(I_q)$
2: $\{F_q^{(j)}\} \leftarrow E(I_q; \boldsymbol{\theta}) \circ \{\boldsymbol{x}^{(j)}\}$
3: $\{\boldsymbol{o}^{(i)}\} \leftarrow E(I_r; \boldsymbol{\theta}) \circ \{\langle \boldsymbol{p}^{(i)} \rangle\}$
4: $\boldsymbol{T} \leftarrow \boldsymbol{T}^{(0)}$
5: **for** $n$ from 0 to $N_{\max}$ **do**
6: $\quad p \leftarrow \text{Scheduler}(n)$
7: $\quad \tilde{f}_q \leftarrow G(\{F_q^{(j)}\}; p)$
8: $\quad \{\tilde{\boldsymbol{p}}^{(i)}\} \leftarrow \{\boldsymbol{T}\boldsymbol{p}^{(i)}\}$
9: $\quad \boldsymbol{\delta} \leftarrow \Delta_{GN}(\tilde{f}_q \circ \{\langle \tilde{\boldsymbol{p}}^{(i)} \rangle\} - \{\boldsymbol{o}^{(i)}\})$
10: $\quad \boldsymbol{T} \leftarrow \boldsymbol{T} \boxminus \boldsymbol{\delta}$
11: **end for**
12: **return** T

$\langle \cdot \rangle$ denotes 2D projection.

Algorithm 1: Image alignment.

Figure 4: **Robustness to initialization.** Our closed-form solution exhibits significantly greater robustness compared to PixLoc.

$\tilde{J}_q$ using Eqs. (17) and (18). The sparse set of 3D points $\{\boldsymbol{p}^{(i)}\}$ is projected onto the reference image to obtain reference descriptors. We obtain feature descriptors for these points using the same feature embedding network $E(\,\cdot\,; \boldsymbol{\theta})$. Starting with the initial camera pose $\boldsymbol{T}^{(0)} \in \mathbf{SE}(3)$, at every iteration we perform Gauss-Newton steps minimizing the residuals $r^{(i)}$, *w. r. t.* the camera transform $\boldsymbol{T}$:

$$r^{(i)}(\boldsymbol{T}) := F_r^{(i)} - \tilde{f}_q(\langle \boldsymbol{T}\boldsymbol{p}^{(i)} \rangle). \tag{24}$$

$p(\boldsymbol{\epsilon})$ explicitly controls the basin of convergence. In practice, we can use any distribution with high initial variance based on the inaccuracy assumed in the initial point projections. As our estimates improve over the course of optimization, we gradually decrease the variance. Notably, the choice of $p(\boldsymbol{\epsilon})$ lets us define values for points outside of the image boundaries. Fig. 2 illustrates examples of the feature maps $\tilde{f}_q$ with $p$ following a Gaussian distribution of increasing variance.

**Comparison to previous work.** Our derivation is agnostic to the underlying feature extractor and can operate with off-the-shelf feature descriptors. An interesting property of the derived $\tilde{f}_q$ is that it enables *continuous* coarse-to-fine alignment. By adjusting the noise prior $p$ in the scheduler (*cf.* Algorithm 1), we can control the basin of convergence dynamically at runtime. We can adjust the distribution $p$ either by choosing a different parameter set of a pre-defined distribution, or by changing the distribution itself to another family. In practice, we start the optimization with a uniform distribution and then switch to the Gaussian distribution with a slowly decreasing variance.

A uniform distribution with a wide support offers a large basin of convergence, which sacrifices accuracy for robustness. The ensuing Gaussian distribution with decreasing variance refines the pose and leads to a more accurate solution. Appendix E provides implementation details of the scheduler.

**Connection between feature matching and featuremetric image alignment.** Note that the computation of $\tilde{f}_q$ using Eq. (17) only depends on the feature descriptors of the interest points in the query image. This observation suggests an interesting interpretation of minimizing the residual in Eq. (24) using $\tilde{f}_q$ with Gauss-Newton optimization as *feature matching*. The Gauss-Newton step is fully determined by the neighboring points of interest. Consequently, it implies that the effectiveness of such optimization-based methods may be limited in comparison to alternative optimization-inspired approaches, which learn the optimization step (Teed and Deng, 2020).

## 6 EXPERIMENTS

**Datasets.** We evaluate our approach on two most popular datasets for large-scale image localization, namely the Aachen Day-Night dataset (Sattler et al., 2018), extended CMU seasons (Toft et al., 2022) and 7Scenes dataset (Shotton et al., 2013). Aachen Day-Night consists of 98 night and 824 day query

Table 1: **Camera localization on Aachen Day-Night and CMU Seasons.** Our evaluation shows an overall improved accuracy in diverse scenarios despite using *self-supervised* SuperPoint descriptors. We compare to the state-of-the-art methods *supervised* with pose: GN-Net (von Stumberg et al., 2020a), LM-Reloc (von Stumberg et al., 2020b) and PixLoc (Sarlin et al., 2021).

| Method | Aachen | | CMU Seasons | | |
| --- | --- | --- | --- | --- | --- |
| | Day | Night | Urban | Suburban | Park |
| GN-Net | 62.4 / 69.4 / 76.9 | 49.0 / **58.2** / 66.3 | 75.4 / 79.9 / 92.6 | 64.7 / 67.0 / 81.4 | 46.7 / 48.3 / 65.3 |
| LM-Reloc | 60.4 / 68.0 / 76.3 | 37.8 / 46.9 / 59.2 | 76.6 / 82.5 / 93.4 | 67.3 / 72.0 / 82.8 | 49.1 / 53.4 / 66.9 |
| PixLoc | 64.3 / 69.3 / 77.4 | **51.0** / 55.1 / **67.3** | **88.3** / 90.4 / 93.7 | 79.6 / 81.1 / 85.2 | 61.0 / 62.5 / 69.4 |
| Ours (SuperPoint) | **66.3** / **72.5** / **78.8** | 43.9 / 50.0 / 56.1 | 86.0 / **90.6** / **95.2** | **79.8** / **85.0** / **92.4** | **63.4** / **67.9** / **77.5** |

images. Extended CMU seasons consists of 14 slices, 5 for urban environment, 5 for suburban and 4 for park. Each slice contains between 3000 and 5000 query images. The 7Scenes dataset comprises seven distinct scenes, each containing multiple sequences of 500 to 1000 frames. For our ablation study in Appendix D we used Cambridge Landmarks dataset (Kendall et al., 2016).

**Results.** Table 1 presents the results for camera localization on Aachen Day-Night (Sattler et al., 2018) and the extended CMU Seasons (Toft et al., 2022). We use SuperPoint (DeTone et al., 2018) as the feature descriptor in these experiments. In each setting (Day/Night for Aachen; Urban/Suburban/Park for CMU Seasons), we report three numbers, which indicate the percentage of the query images that were successfully localized within the specified translation and rotation thresholds. We adopt the standard threshold values defined by the benchmarks (translation, rotation): (0.25m, 2°) / (0.5m, 5°) / (5m, 10°). We adopted the implementation from PixLoc (Sarlin et al., 2021) to run these benchmarks.[1] A comparison with PixLoc on the 7Scenes dataset is presented in Appendix F.

We observe that our solution, despite using self-supervised descriptors from SuperPoint (DeTone et al., 2018), achieves an overall strong localization accuracy across diverse settings in comparison to supervised frameworks. Aachen Day-Night has significant occlusions, hence many outliers. Since the Gauss-Newton loss does not incorporate any outlier filtering, we do not expect high accuracy for our approach on this dataset. Nevertheless, we were surprised to find that on Aachen Day our approach even slightly surpassed previous state of the art. On Aachen Night, our accuracy is inferior to previous work, however. This is somewhat expected, since SuperPoint feature descriptors were not directly trained on day-night correspondences. By contrast, PixLoc was trained with supervision on day-night image pairs, which provides an obvious advantage. On CMU Seasons, our approach demonstrates a clear improvement over previous featuremetric alignment methods.

The proposed approach demonstrates notable robustness to initialization noise. We compared the proposed scheme and PixLoc on the Cambridge Landmarks dataset, varying the levels of random noise applied to the ground-truth pose as illustrated in Fig. 4 and Appendix G.

Overall, these results empirically confirm the validity of our closed-form derivation. Furthermore, the comparison suggests a limited benefit of end-to-end learning with the Gauss-Newton loss. By contrast, a dynamic convergence basin offered by our analytical solution provides versatility *w. r. t.* the underlying feature descriptors and improves robustness to pose initialization, as a result.

## 7 CONCLUSION

We derived a closed-form solution to the Gauss-Newton loss in the context of direct image alignment, which offers two main advantages. First, it allows for dynamic control of the convergence basin, which improves robustness of the alignment to pose initialization. Furthermore, despite using self-supervised descriptors, such control leads to compelling accuracy of pose estimates in comparison to supervised pipelines on established benchmarks. Second, our derivation exposes intrinsic limitations of employing the Gauss-Newton loss in deep learning, as it only leads to a form of interpolation between the feature descriptors in the interest points. This insight offers an interesting connection between direct image alignment and feature matching, and leads to a novel perspective on learning robust features end-to-end, which we will investigate in future work.

---

[1]https://github.com/cvg/pixloc

## ACKNOWLEDGMENTS

This work was supported by the ERC Advanced Grant SIMULACRON and by TUM Georg Nemetschek Institute under the project AI4TWINNING. We thank Linus Härenstam-Nielsen for helpful discussions and proofreading.

## REPRODUCIBILITY STATEMENT

The main contribution of this work is the derivation of the closed-form solution to the Gauss-Newton loss. We provide details of this derivation in Appendix B. To reproduce our experimental results, we elaborate on the implementation details Appendix E. To facilitate reproducibility in future research, we also publicly release our code.

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

## A  OPTICAL FLOW BIAS

We demonstrate that PixLoc exhibits bias inherited from the training dataset. Fig. 5 illustrates the average translation error after optimization when evaluated on the Cambridge dataset, highlighting the impact of different initial displacements along the image axes. To solely evaluate the bias learned by the feature maps, we disabled dataset-specific learned movement priors denoted as $\lambda$ in PixLoc (Sarlin et al., 2021). Our results demonstrate that when trained on the CMU dataset, the network predicts optical flow along the X axis more accurately. This increased accuracy is directly linked to the structure of the CMU dataset. Specifically, the dataset comprises image pairs captured by a camera inside a moving car, pointed towards the side of the road. Consequently, this positioning results in the network predominantly learning horizontal optical flow patterns, reflecting the lateral movement observed in the images of the training dataset.

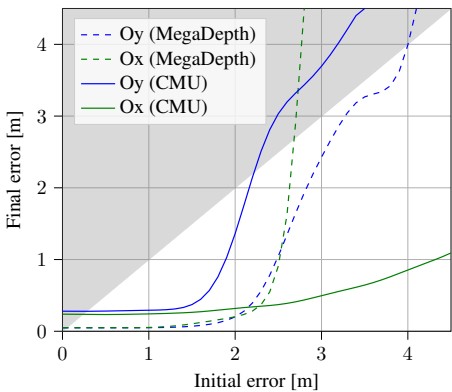

Figure 5: The final error of the PixLoc solution along $X/Y$- axes *w. r. t.* the initial error. PixLoc remains accurate only along $X$-axis on CMU, reflecting the dominant motion in the training set.

## B  DERIVATION OF THE ANALYTICAL SOLUTION

In this section, we provide a solution to minimizing Eq. (16) from the main text. Let $\Omega$ be the coordinate domain of an image plane $[0, 1] \times [0, 1]$, and let $E \subseteq \mathbb{R}^d$ define the feature space. We aim to find $\tilde{f} : \Omega \to E$:

$$\tilde{f} = \arg\min_{f} \sum_{j} \int_{\Omega} \left\| \tilde{\boldsymbol{x}} - (J(\tilde{\boldsymbol{x}})^{\intercal} J(\tilde{\boldsymbol{x}}))^{-1} J(\tilde{\boldsymbol{x}})^{\intercal} \left( f(\tilde{\boldsymbol{x}}) - F_q^{(j)} \right) - \boldsymbol{x}^{(j)} \right\|_2^2 p(\tilde{\boldsymbol{x}} - \boldsymbol{x}^{(j)}) d\tilde{\boldsymbol{x}}. \quad (25)$$

We start by relaxing the constraint $\nabla f(\boldsymbol{x}) = J(\boldsymbol{x})$. This makes every value of $f(\boldsymbol{x})$ and $J(\boldsymbol{x})$ locally independent and, therefore, the minimum is achieved by minimizing Eq. (25) independently over every point in $\Omega$:

$$\{\tilde{f}(\tilde{\boldsymbol{x}}), \tilde{J}(\tilde{\boldsymbol{x}})\} = \arg\min_{\hat{\boldsymbol{f}}, \hat{\boldsymbol{J}}} \sum_{j} \left\| \tilde{\boldsymbol{x}} - \left( \hat{\boldsymbol{J}}^{\intercal} \hat{\boldsymbol{J}} \right)^{-1} \hat{\boldsymbol{J}}^{\intercal} \left( \hat{\boldsymbol{f}} - F_q^{(j)} \right) - \boldsymbol{x}^{(j)} \right\|_2^2 p(\tilde{\boldsymbol{x}} - \boldsymbol{x}^{(j)}). \quad (26)$$

Note that the only significant part of $\hat{\boldsymbol{f}}$ is a part which is spanned by columns of $\hat{\boldsymbol{J}}$. This can be easily seen by observing that $\hat{\boldsymbol{J}}^{\intercal} \hat{\boldsymbol{f}}$ is a non-normalized projection onto $\hat{\boldsymbol{J}}$. More formally, let us find $\hat{\boldsymbol{f}}$ as

$\hat{J}a + \hat{J}^\perp b$, where $\hat{J}^\perp$ is a basis for an orthogonal complement for $\mathrm{span}(\hat{J})$. By substituting this into Eq. (26), we obtain:

$$\{\tilde{a}(\tilde{x}), \tilde{J}(\tilde{x})\} = \underset{a, \hat{J}}{\arg\min} \sum_j \left\| \tilde{x} - x^{(j)} - a + \left(\hat{J}^\intercal \hat{J}\right)^{-1} \hat{J}^\intercal F_q^{(j)} \right\|_2^2 p(\tilde{x} - x^{(j)}), \tag{27}$$

$$\tilde{f}(\tilde{x}) \in \left\{ \tilde{J}(\tilde{x})\tilde{a}(\tilde{x}) + \tilde{J}(\tilde{x})^\perp b \,|\, b \in \mathbb{R}^{d-2} \right\}.$$

Observe that this problem is a linear least squares in $a$ and $\left(\hat{J}^\intercal \hat{J}\right)^{-1} \hat{J}^\intercal$ and recall that $p(\cdot) \geq 0$ (by definition), hence the problem is convex. Minimality conditions for Eq. (27) for $a$ and $\left(\hat{J}^\intercal \hat{J}\right)^{-1} \hat{J}^\intercal$ are:

$$\frac{d}{da}: \quad \sum_j \left( \tilde{x} - x^{(j)} - a + \left(\hat{J}^\intercal \hat{J}\right)^{-1} \hat{J}^\intercal F_q^{(j)} \right) p(\tilde{x} - x^{(j)}) = 0, \tag{28}$$

$$\frac{d}{d\left(\hat{J}^\intercal \hat{J}\right)^{-1} \hat{J}^\intercal}: \quad \sum_j \left( \tilde{x} - x^{(j)} - a + \left(\hat{J}^\intercal \hat{J}\right)^{-1} \hat{J}^\intercal F_q^{(j)} \right) F_q^{(j)\intercal} p(\tilde{x} - x^{(j)}) = 0. \tag{29}$$

From Eq. (28), we have:

$$a = \tilde{x} - \mathbf{x}_m(\tilde{x}) + \left(\hat{J}^\intercal \hat{J}\right)^{-1} \hat{J}^\intercal \mathbf{y}_m(\tilde{x}), \tag{30}$$

where

$$\mathbf{y}_m(\tilde{x}) := \sum_j F_q^{(j)} p(\tilde{x} - x^{(j)}) / \sum_j p(\tilde{x} - x^{(j)}),$$

$$\mathbf{x}_m(\tilde{x}) := \sum_j x^{(j)} p(\tilde{x} - x^{(j)}) / \sum_j p(\tilde{x} - x^{(j)}). \tag{31}$$

Substituting $a$ into Eq. (29), we obtain

$$\sum_j \left( \mathbf{x}_m(\tilde{x}) - x^{(j)} + \left(\hat{J}^\intercal \hat{J}\right)^{-1} \hat{J}^\intercal \left( F_q^{(j)} - \mathbf{y}_m(\tilde{x}) \right) \right) F_q^{(j)\intercal} p(\tilde{x} - x^{(j)}) = 0, \tag{32}$$

$$-\sum_j \left( x^{(j)} - \mathbf{x}_m(\tilde{x}) \right) F_q^{(j)\intercal} p(\tilde{x} - x^{(j)}) +$$

$$\left(\hat{J}^\intercal \hat{J}\right)^{-1} \hat{J}^\intercal \sum_i \left( F_q^{(j)} - \mathbf{y}_m(\tilde{x}) \right) F_q^{(j)\intercal} p(\tilde{x} - x^{(j)}) = 0, \tag{33}$$

$$-\mathrm{Cov}_{\mathbf{xy}}(\tilde{x}) + \left(\hat{J}^\intercal \hat{J}\right)^{-1} \hat{J}^\intercal \mathrm{Cov}_{\mathbf{y}}(\tilde{x}) = 0, \tag{34}$$

where

$$\mathrm{Cov}_{\mathbf{xy}}(\tilde{x}) := \sum_j \left( x^{(j)} - \mathbf{x}_m(\tilde{x}) \right) \left( F_q^{(j)} - \mathbf{y}_m(\tilde{x}) \right)^\top p(\tilde{x} - x^{(j)}), \tag{35}$$

$$\mathrm{Cov}_{\mathbf{y}}(\tilde{x}) := \sum_j \left( F_q^{(j)} - \mathbf{y}_m(\tilde{x}) \right) \left( F_q^{(j)} - \mathbf{y}_m(\tilde{x}) \right)^\top p(\tilde{x} - x^{(j)}). \tag{36}$$

Solving Eq. (34) for $J$ and substituting the solution into Eq. (27) results in

$$\tilde{J}(\tilde{x}) = \left( \mathrm{Cov}_{\mathbf{xy}}(\tilde{x}) \mathrm{Cov}_{\mathbf{y}}(\tilde{x})^{-1} \right)^\dagger, \tag{37}$$

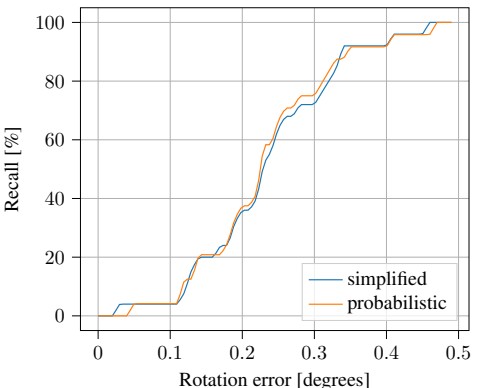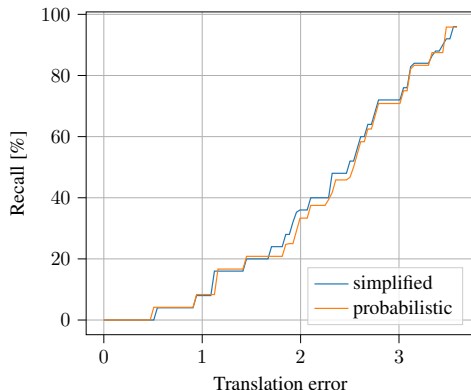

Figure 6: Comparison of rotation and translation recalls for our simplified version of GN-loss and probabilistic one

$$\tilde{f}(\tilde{\boldsymbol{x}}) = \tilde{J}(\tilde{\boldsymbol{x}})\boldsymbol{a} = \tilde{J}(\tilde{\boldsymbol{x}})\big(\tilde{\boldsymbol{x}} - \mathbf{x}_m(\tilde{\boldsymbol{x}})\big) + \tilde{J}(\tilde{\boldsymbol{x}})\left(\tilde{J}(\tilde{\boldsymbol{x}})^\mathsf{T}\tilde{J}(\tilde{\boldsymbol{x}})\right)^{-1}\tilde{J}(\tilde{\boldsymbol{x}})^\mathsf{T}\mathbf{y}_m(\tilde{\boldsymbol{x}}). \tag{38}$$

$\tilde{J}(\tilde{\boldsymbol{x}})\left(\tilde{J}(\tilde{\boldsymbol{x}})^\mathsf{T}\tilde{J}(\tilde{\boldsymbol{x}})\right)^{-1}\tilde{J}(\tilde{\boldsymbol{x}})^\mathsf{T}\mathbf{y}_m(\tilde{\boldsymbol{x}})$ can be simplified. Note that in the Gauss-Newton step we are projecting $\tilde{f}(\tilde{\boldsymbol{x}})$ onto $\tilde{J}(\tilde{\boldsymbol{x}})$. $\tilde{J}(\tilde{\boldsymbol{x}})\left(\tilde{J}(\tilde{\boldsymbol{x}})^\mathsf{T}\tilde{J}(\tilde{\boldsymbol{x}})\right)^{-1}\tilde{J}(\tilde{\boldsymbol{x}})^\mathsf{T}$ is a projection operator, so we are projecting two times. Therefore, there is another equivalent solution which we will further use for the sake of simplicity:

$$\tilde{f}(\tilde{\boldsymbol{x}}) = \tilde{J}(\tilde{\boldsymbol{x}})\big(\tilde{\boldsymbol{x}} - \mathbf{x}_m(\tilde{\boldsymbol{x}})\big) + \mathbf{y}_m(\tilde{\boldsymbol{x}}). \tag{39}$$

Observe that substituting $\tilde{f}$ from Eq. (39) and Eq. (38) into Eq. (26) yields the same loss value. Notably, $\boldsymbol{\nabla}\tilde{f}(\tilde{\boldsymbol{x}}) = \tilde{J}(\tilde{\boldsymbol{x}})$ for a uniform distribution $p(\cdot)$.

## C  PROBABILISTIC *vs.* SIMPLIFIED GAUSS-NEWTON LOSS

In this section, we compare our simplified Gauss-Newton loss formulation,

$$\underset{\hat{\boldsymbol{f}},\hat{\boldsymbol{J}}}{\arg\min}\sum_j \left\| \tilde{\boldsymbol{x}} - \left(\hat{\boldsymbol{J}}^\mathsf{T}\hat{\boldsymbol{J}}\right)^{-1}\hat{\boldsymbol{J}}^\mathsf{T}\left(\hat{\boldsymbol{f}} - F_q^{(j)}\right) - \boldsymbol{x}^{(j)} \right\|_2^2 p(\tilde{\boldsymbol{x}} - \boldsymbol{x}^{(j)}), \tag{40}$$

to the original loss, which was derived through the maximum likelihood estimation (MLE),

$$\underset{\hat{\boldsymbol{f}},\hat{\boldsymbol{J}}}{\arg\min}\sum_j \left( \left\| \hat{\boldsymbol{J}}\left(\tilde{\boldsymbol{x}} - \left(\hat{\boldsymbol{J}}^\mathsf{T}\hat{\boldsymbol{J}}\right)^{-1}\hat{\boldsymbol{J}}^\mathsf{T}\left(\hat{\boldsymbol{f}} - F_q^{(j)}\right) - \boldsymbol{x}^{(j)}\right) \right\|_2^2 - \mathrm{logdet}(\hat{\boldsymbol{J}}^\mathsf{T}\hat{\boldsymbol{J}}) \right) p(\tilde{\boldsymbol{x}} - \boldsymbol{x}^{(j)}). \tag{41}$$

We note that the solution to the probabilistic loss is much more complicated and computationally expensive. Derivatives for this formulation are high-order polynomials in elements of $\hat{\boldsymbol{f}}$ and $\hat{\boldsymbol{J}}$. It could be solved by general methods of algebraic geometry like Gröbner basis or Homotopy continuation, but the existence of a closed-form solution is not guaranteed, and we were not able to find one.

Nevertheless, in order to compare the two formulations, we approached the problem numerically. Fig. 6 presents the results. We used the Aachen dataset with hloc poses to plot the recall of query

images as a function of rotation and translation errors. We observe that the accuracy difference between these two formulations is negligible in practice. However, the simplified version admits a closed-form solution and can be computed several orders of magnitudes more efficiently.

## D  A STUDY OF FEATURE DESCRIPTORS

We show that our approach is agnostic to the choice of the underlying feature descriptor. To analyze the accuracy of our algorithm in terms of the localization error, we use Cambridge Landmarks (Kendall et al., 2016), which provides ground-truth poses.

We experiment with four popular feature descriptors: SIFT (Lowe, 2004), SOLD2 (Pautrat et al., 2021), SuperPoint (DeTone et al., 2018) and the recent SiLK (Gleize et al., 2023). We plug them in as the embeddings for interest points detected by SuperPoint. Fig. 7 plots recall (the percentage of successfully localized queries) as a function of the tolerated translation error.

We observe that SuperPoint outperforms all other feature descriptors. SOLD2 (Pautrat et al., 2021) is substantially more inferior in terms of accuracy of the pose estimates. Since it is designed to be a line descriptor, its representation of non-linear structures is not well-defined, thus such outcome is somewhat expected. Interestingly, SuperPoint and SIFT both surpass the more recent SiLK on this benchmark. In comparison to SuperPoint, SiLK has 128-*D* descriptor, which suggests that it may be less expressive than the 256-*D* SuperPoint descriptors.

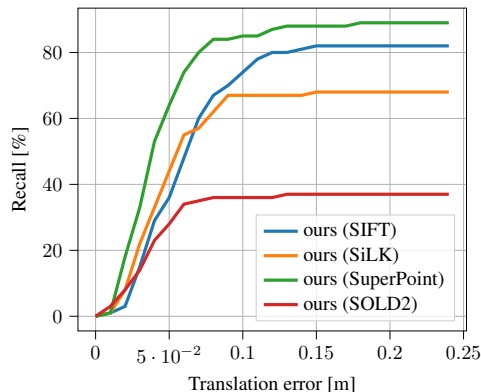

Figure 7: We study our approach with different feature descriptors on Cambridge Landmarks.

## E  IMPLEMENTATION DETAILS.

We implement our approach in PyTorch (Paszke et al., 2019). The primary focus of our implementation is the experimental validation of our derived results. Therefore, we did not optimize the runtime performance, which heavily depends on the number of interest points and 3D points involved in the computation. On average, the code takes approximately 6 seconds and 10 seconds per alignment on CMU and Aachen, respectively, on a single NVIDIA A4000. The scheduler in Algorithm 1 adapts the distribution $p(\cdot)$ as follows. We initialize $p(\cdot)$ with a truncated uniform distribution of a fixed radius around all interest points. The radius decreases from $50\%$ of the image diagonal to $5\%$ in the first 30 iterations. Afterward, the scheduler switches to the normal distribution around each interest point with standard deviation $\sigma$. Initially, we define $\sigma$ such that $99\%$ of the distribution covers $10\%$ of the image around the point, and we decrease the coverage ratio to $1\%$.

Since our formulation does not have any outlier removal, we adopt the cut-off from the coarse tracker of DSO (Engel et al., 2016). The idea is to define a threshold that discards all but $20\%$ of the residuals with the lowest norm. In our experiments with SuperPoint, we set the threshold value to 0.4.

## F  EVALUATION ON 7SCENES

Here, we complement our evaluation with the 7Scenes dataset. This dataset features substantial blur and distortions in some scenes, as it was captured with a rolling-shutter Kinect camera. Although SuperPoint features were not trained with such distortions, direct alignment with our closed-form solution remains competitive with PixLoc, both in terms of the median error and the recall Table 2.

Table 2: **7Scenes (Shotton et al., 2013) evaluation and comparison to PixLoc.** We report the median rotation and translation errors, as well as recall values at the specified thresholds of translation and rotation.

| Scene | Method | Median error | Recall | | | |
|---|---|---|---|---|---|---|
| | | | (1cm,1°) | (5cm,5°) | (25cm,2°) | (50cm,5°) |
| Heads | Ours | **0.013m**, 1.006° | 24.80% | **93.0%** | **86.30%** | **92.0%** |
| | Pixloc | **0.013m**, **0.863°** | **36.40%** | 85.60 % | 84.00% | 85.90% |
| Office | Ours | 0.028m, 0.941° | 6.12 % | 80.25% | 92.55% | **99.00%** |
| | Pixloc | **0.026m**, **0.792°** | **7.95%** | **80.70 %** | **93.12 %** | 96.85% |
| Redkitchen | Ours | 0.037m, 1.444° | 1.56 % | 64.42% | 71.88% | **90.16%** |
| | Pixloc | **0.034m**, **1.217°** | **3.74%** | **67.78 %** | **76.56 %** | 89.48% |
| Pumpkin | Ours | 0.049m, 1.555° | 1.60 % | 51.65% | 62.65% | **85.35%** |
| | Pixloc | **0.041m**, **1.173°** | **2.80%** | **59.75 %** | **71.00 %** | 84.40% |
| Stairs | Ours | 0.154m, 3.685° | 1.90 % | 19.90 % | 25.60 % | 60.50 % |
| | Pixloc | **0.048m**, **1.268°** | **2.60%** | **51.10 %** | **59.10 %** | **74.70%** |
| Chess | Ours | 0.026m, 0.904° | 5.50 % | **91.95%** | **95.45%** | **98.45%** |
| | Pixloc | **0.024m**, **0.812°** | **7.75%** | 90.75 % | 94.95% | 96.15% |
| Fire | Ours | 0.021m, 0.978° | 10.40 % | **90.65%** | **94.05%** | **98.45%** |
| | Pixloc | **0.019m**, **0.781°** | **15.85%** | 87.50 % | 87.20% | 90.10% |

## G    QUALITATIVE EXAMPLES

Fig. 8 visualizes convergence examples of one scene with different pose initializations. The examples consistently demonstrate successful alignment despite suboptimal initialization of varying degree.

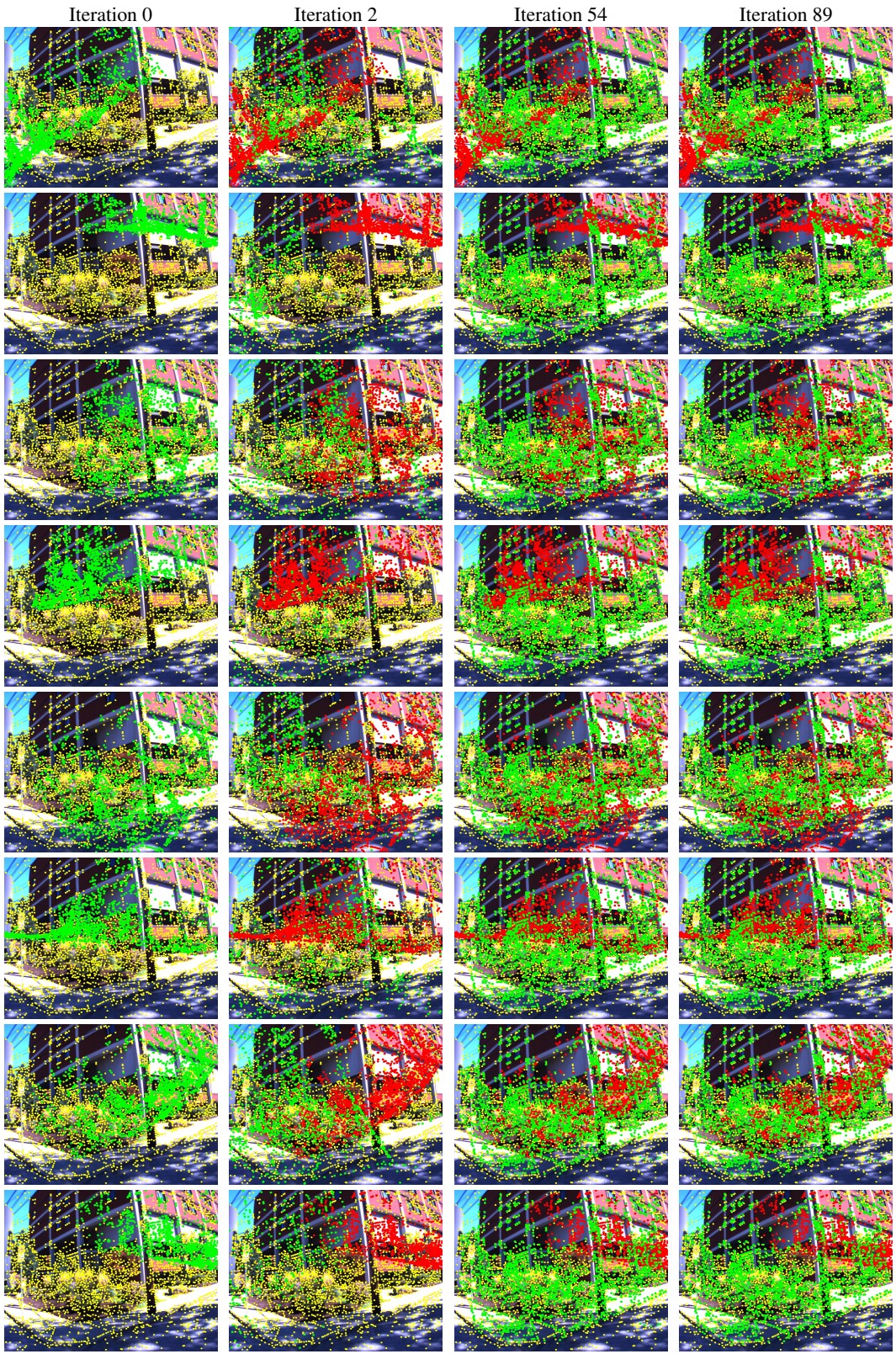

Figure 8: Examples of convergence from random initial poses. The green points are projections of 3D points using the current pose estimate; the red points are projections of the 3D points with the initial pose, and the yellow points denote the locations of interest points. It can be seen that despite these highly inaccurate initial poses, our approach converges to the correct solution.

