# OpenReview forum: "An Analytical Solution to Gauss-Newton Loss for Direct Image Alignment"
_ICLR.cc/2024/Conference — ICLR 2024 oral_

### Official Review · Reviewer_1Yhm · 2023-10-30

**Soundness:** 3 good
**Presentation:** 3 good
**Contribution:** 3 good
**Rating:** 8
**Confidence:** 3

**Summary:**

This paper builds on the Gauss-Newton loss and establishes a closed-form solution for the expected optimum of this loss; it doesn't depend on the specific feature representation being used, and it enables the adjustment of the convergence basin based on assumptions about the uncertainty in the current estimates. This provides a means to effectively control the convergence properties of the algorithm. Notably, even when employing self-supervised feature embeddings, this approach attains impressive accuracy compared to the SOTA direct image alignment methods that are trained end-to-end with pose supervision. Furthermore, it demonstrates enhanced robustness in terms of pose initialization.

**Strengths:**

To the best of my knowledge, the closed-form derivative of the Gauss-Newton loss is innovative, and its effectiveness has been confirmed through empirical evaluation within the domain of direct image alignment, specifically with self-supervised feature descriptors - SuperPoint. What's particularly noteworthy is that this derivative can be applied to other areas to encompass methods employing backpropagation through Gauss-Newton or Levenberg-Marquardt optimization, among others.

**Weaknesses:**

No major weakness.
1. It would be interesting to see more discussions on the insight to the end-to-end learning framework's limitation, and a solution to that.
2. It would be interesting to see this approach handles outliers inherently.
3. It would be interesting to see this approach is applied to other areas.

**Questions:**

N/A

---

> ### Author Response · Authors · 2023-11-16
>
> Dear Reviewer,
>
> Thank you for your insightful suggestions and positive review. We are glad to hear that you consider our contribution innovative.
>
> Regarding the points raised in the weaknesses section, they indeed present interesting directions for further research. We are currently exploring the feasibility of end-to-end network training with our closed-form solution. In our preliminary experiments, we have found that backpropagation through this solution poses challenges due to numerical instability, and we are actively investigating potential solutions to this issue.
>
> As we explain in the Implementation Details appendix, we currently address outliers only at the feature alignment stage. To manage outliers prior to the feature map creation stage (on the point of interest’s level), information from both reference and target images is required. However, our map creation process is based solely on a single image.
>
> We are also excited about the prospect of applying our method in other domains.
>
> Once again, thank you for your valuable feedback and support.

---

### Official Review · Reviewer_TZP4 · 2023-10-30

**Soundness:** 3 good
**Presentation:** 3 good
**Contribution:** 3 good
**Rating:** 6
**Confidence:** 3

**Summary:**

The authors propose a closed-form solution to the Gauss-Newton loss in the field of direct image alignment. This method allows for dynamic control of the convergence basin to improve the robustness of the alignment to pose initialization. Moreover, the proposed method shows the intrinsic limitations of employing Gauss-Newton loss in deep learning, which offers an insight between direct image alignment and feature matching. The simulation experiments have shown its superior performance.

**Strengths:**

1.	The paper provides an analytical solution to the Gauss-Newton loss, which is a novel technology for generating a dense feature map.
2.	The paper shows the inherent limitations of feature learning with backpropagation via the Gauss-Netwon optimization.
3.	The paper is well-organized and shows the explicit introduction to notion of the Gauss-Newton.

**Weaknesses:**

1.	The paper is required to give more comparisons with state-of-the-art in terms of accuracy of SE3
2.	Can the authors provide more training details of the proposed method, for example, the feature embedding network E, the learning rate, the batch size.

**Questions:**

See the weakness part

---

> ### Author Response · Authors · 2023-11-16
>
> Dear Reviewer,
>
> Thank you for your valuable review and for acknowledging the novel aspects of our work.
>
> In response to your first point about comparing our method with state-of-the-art methods in terms of SE3 accuracy, we have conducted extensive evaluations across various well-known datasets. Our comparisons, as detailed in the paper (e.g., in Table 1), focus on the Recall with respect to translation and rotation thresholds, which directly reflect the accuracy of the estimated SE(3) camera pose. If there are specific datasets or aspects that you believe we may have overlooked, we would appreciate your guidance to further enhance our comparative analysis.
>
> Regarding your second point on the absence of training details, our method does not involve a traditional training process. However, it does leverage self-supervised SuperPoint descriptors. We will make it clearer in the updated manuscript.
>
> Thank you once again for your valuable feedback.

---

> > ### Comment · Reviewer_TZP4 · 2023-11-20
> >
> > Thanks. The authors have addressed my concerns.

---

### Official Review · Reviewer_hDb9 · 2023-10-31

**Soundness:** 3 good
**Presentation:** 4 excellent
**Contribution:** 3 good
**Rating:** 8
**Confidence:** 3

**Summary:**

This paper addresses the task of Direct Image Alignment, which is used to estimate the relative 6DoF pose between two images. The task is strongly affected by pose initialization, which has been addressed by prior art by switching to optimization methods that increase the convergence basin, such as the Gauss-Newton loss. The authors claim that these prior methods induce bias towards the training data which limits their generalization.
The papers main contribution addresses this problem. The authors introduce an analytical close from solution to the Gauss-Newton loss. This solution is independent of the feature representation and enables adjustment of the convergence basin based on the uncertainty in current estimates, giving control over the algorithm’s convergence properties. This property is used during the experimental evaluation, where optimization is first performed on a uniform distribution with a wider range, but then is switched out to a Gaussian with an increasingly narrowing distribution.
Their secondary contributions are insights that the analytical solution provides. Specifically, they show that under their simplified conditions, the Gauss-Newton step is determined by the neighboring points of interest. The author conclude that this is inherently limiting in comparison to other optimization methods.
Experimental results demonstrate superior performance in almost all results over supervised state-of-the-art methods using self-supervised descriptors.
The appendix provides further insights on the derivations, as well as more interesting experimental results.

**Strengths:**

1) Well-written paper. It was a joy to read. It explains the context of the problem well, as well as establishing the necessary preliminary knowledge before delving into its actual contribution. There is a minor exception to this for the derivations (see Weaknesses)
2) Under the simplifying assumption that eps follows an isotropic Gaussian, the authors derive a close-form solution to the minimizer of the Gauss-Netwon loss in expectation. Under the assumption that the authors claim about poor generalization due to training-data-biased feature maps holds (see Weaknesses), the proposed solutions has the main advantage that it provides unbiased feature map. In addition, it provides the ability to control the basin of convergence, which in turn makes the proposed method more robust to bad initialization (cf. Fig 3). Lastly, the assumed simplification which was necessary to derive the closed-form solution has been shown to lead to negligible differences (cf. Fig 5)
3) Using self-supervised features, the proposed method is capable of outperforming supervised related work on almost all metrics. This is a strong statement, as the method can be used in conjunction with large and powerful foundation models, enabling bigger generalization due to the superior dataset sizes of such models. Therefore it is complimentary to these works.
4) The authors provide an interesting insight when using Gauss-Newton as feature matching and indicate that it may be inherently limited. This is important for informing future work in optimization-based methods. Further analysis also shows that joint training of both losses for L_GN may lead to numerical instability and may shed light on reported training divergence of prior work.

**Weaknesses:**

1) My biggest gripe with the paper is that their claim that motivates the approach is not empirically validated and there is no mention of such validation elsewhere. The authors claim both in the abstract as well as in the appendix that prior art use feature maps that may embed the inductive bias of the training data. While I can comprehend the underlying reasoning, such a claim needs to be empirically shown.
For example, an experiment on out-of-distribution test sets demonstrating the superiority of the closed-form solution would back the authors claims and in turn strengthen the paper.
2) On the same topic of bias, I argue that the authors should explicitly state that their method still exhibits bias, but that the source of this is the underlying feature representation (result of this can be seen in Tbl. 1, Aachen Night dataset). Otherwise it may read that the authors claim their method is not biased. This is stated at the end of Section 3, but I think it should be stated clearly in either Abstract, Introduction and Conclusion section. This is a minor point however and only serves to improve clarity.
3) A little contradictory to my point in the Strengths section, I believe the math heavy section 4 and 5 could be made a little more clearer when derivations skip multiple steps. Otherwise the sections read as if one equation immediately follows from the other, which is not always the case. (e.g Eq 6. -> Eq. 7). This would enhance the readability of the paper.
4) On the Aachen-Night dataset, the proposed method clearly suffers. The authors claim that this is due to the underlying feature representation used, which was not trained day-night correspondences. While I find the reasoning sound, it would help the authors claim to have used a feature representation that have used such correspondences during training. This in turn would again strengthen the papers contribution and indicate that it can work in different settings.

**Questions:**

Questions:
- Eq. 23 leads to numerical instability. Is there a way to avoid this for stochastic optimization?
- Fig. 4) The median is stable for all tested errors. For what ranges does this hold? I.e How far can the initial error be?
- Fig. 4) Is there a similar plot for translation?

Comments:
- Sec 2, Image Alignment: Add the variable T to make the text more consistent with the rest -> "(...) estimate the relative 6DoF camera pose T."

---

> ### Author Response · Authors · 2023-11-19
>
> Dear Reviewer,
>
> Thank you for your comprehensive review of our paper and for acknowledging its strength.
>
> We have taken your feedback into consideration and have accordingly modified the manuscript to address the key points you raised:
> 1) To empirically support our claim regarding the inductive bias of training data, we have added additional analysis to Appendix A.
> 2) We have refined our claim on the absence of bias in our method (at the end of Sec. 3) to make it more concrete.
> 3)  We modified the transition between Eq. 6 and Eq. 7 to show that there is no direct connection between them.
> 4) We agree that using a feature representation trained on day-night correspondences would be a significant enhancement. Unfortunately, we could not find an off-the-shelf feature embedder trained for the day-night pairs.
> (Note that training such a feature embedder would require supervision.)
>
> Additionally, we have addressed your questions in the revised manuscript:
>
> The revised version of Fig. 4 now contains an extended analysis of the stability of the median error across broader ranges.
>
> We have added a corresponding plot for robustness w.r.t. translation to supplemental material.
>
> We are currently exploring the feasibility of end-to-end network training with our closed-form solution, but addressing all issues lies beyond the scope of this work.
>
> Thank you again for your valuable feedback and for supporting our work.

---

> ### Comment · Reviewer_hDb9 · 2023-11-22
> **Points addressed**
>
> I thank the authors for addressing the points raised in my review. It is a pity that a day-night correspondence feature representation experiment could not be performed, but I understand the authors reasoning.
>
> As I already voted for acceptance, I will keep my original voting.

---

### Meta-Review · Area_Chair_Cmod · 2023-12-10

**Metareview:**

A closed form solution to the expected optimum for Gauss-Newton loss in direct image alignment is proposed, which does not depend on the feature representation and allows adjusting the convergence basin based on current uncertainty. This is an important technical contribution that furthers understanding of direct image alignment, strongly supported by experiments with self-supervised features that outperform pose-supervised methods on standard datasets.

All the reviewers agree that the proposed closed-form solution to the Gauss-Newton loss is interesting and consequential to increased robustness of image alignment. They also agree that the experimental results utilizing only self-supervised descriptors are convincing in comparison to approaches supervised with pose estimates. The connections made to end-to-end learning, as well as between feature matching and direct image alignment also open up potential directions for future research. The AC agrees with the reviewer consensus that the paper may be accepted for presentation at ICLR.

**Justification For Why Not Higher Score:**

Not applicable.

**Justification For Why Not Lower Score:**

An interesting solution to a well-known problem with the potential to form connections to broader areas, supported by a crisp presentation and strong experiments.

---

### Decision · Program_Chairs · 2024-01-16

Accept (oral)